# Once upon a time in Mexico: Holocene biogeography of the spotted bat (*Euderma maculatum*)

Daniel Enrique Sanchez[1,2]*, Faith M. Walker[1,2], Colin J. Sobek[1,2], Cori Lausen[3], Carol L. Chambers[1]

**1** Bat Ecology & Genetics Lab, School of Forestry, Northern Arizona University, Flagstaff, AZ, United States of America, **2** The Pathogen & Microbiome Institute, Northern Arizona University, Flagstaff, AZ, United States of America, **3** Wildlife Conservation Society Canada, Kaslo, British Columbia, Canada

* Daniel.Sanchez@nau.edu

**Data Availability Statement:** DNA sequences (D-loop, cytB) are available in GenBank under the

## Abstract

Holocene-era range expansions are relevant to understanding how a species might respond to the warming and drying climates of today. The harsh conditions of North American deserts have phylogenetically structured desert bat communities but differences in flight capabilities are expected to affect their ability to compete, locate, and use habitat in the face of modern climate change. A highly vagile but data-deficient bat species, the spotted bat (*Euderma maculatum*), is thought to have expanded its range from central Mexico to western Canada during the Holocene. With specimens spanning this latitudinal extent, we examined historical demography, and used ecological niche modeling (ENM) and phylogeography (mitochondrial DNA), to investigate historic biogeography from the rear to leading edges of the species' range. The ENM supported the notion that Mexico was largely the Pleistocene-era range, whereas haplotype pattern and Skyline plots indicated that populations expanded from the southwestern US throughout the Holocene. This era provided substantial gains in suitable climate space and likely facilitated access to roosting habitat throughout the US Intermountain West. Incongruent phylogenies among different methods prevented a precise understanding of colonization history. However, isolation at the southern-most margin of the range suggests a population was left behind in Mexico as climate space contracted and are currently of unknown status. The species appears historically suited to follow shifts in climate space but differences in flight behaviors between leading edge and core-range haplogroups suggest range expansions could be influenced by differences in habitat quality or climate (e.g., drought). Although its vagility could facilitate response to environmental change and thereby avoid extinction, anthropogenic pressures at the core range could still threaten the ability for beneficial alleles to expand into the leading edge.

## Introduction

Modern-day climate change is an existing and threatening influence on the distribution of biodiversity [1, 2]. Western North America is becoming warmer and drier [3, 4], a trend that is

following accession numbers: OP234456, OP234511.

**Funding:** The authors received no specific funding for this work.

**Competing interests:** The authors have declared that no competing interests exist.

already shifting the trajectories of ecosystems in North America [5]. Globally, species are responding to warming climates by expanding into northern latitudes or into higher elevations [6]. Species unable to expand their ranges must rely on phenotypic plasticity or retain enough genetic diversity to adapt to changes in their environment [7, 8]. The more prominent effects of climate change occur on the leading and rear edges of a species distribution [9], which are often observed as a latitudinal (polar) orientation in North America. Leading edge populations are positioned on the frontline of environmental change and may live in the colder extremes of their physiology [10]. Rear edge populations may be either stable or trailing [11]. Trailing rear edge populations endure warmer extremes and are likely to face extirpation. Stable rear edge populations persist in the warmest extremes and are substantially isolated from the core distribution. Increasingly, models are built to predict future distributions of a species by using contemporary geographic occurrences [12] and are attractive methods for predicting the response of a species to climate change. Yet suitability maps of shifting climate alone do not provide insight into how the species might respond to those shifts. The events of the Pleistocene and Holocene provide the most recent frame of reference for extreme climate change and phylogeographic analysis at the transition of those eras can provide important insight into how a species may respond in the future [13].

Bats are remarkable indicators of climate change due to their powered flight [14]. In recent decades, a common and relatively sedentary bat of the Mediterranean, Kuhl's pipistrelle (*Pipistrellus kuhlii)*, has consistently expanded its range into northeastern Europe in response to warming winter temperatures in the upper latitudes [15]. Bats of the harsh deserts of western North America provide a relevant arena of study for range expansions. Over species-level time scales these bat communities are thought to have been structured by habitat filtering via their harsh, arid environments [16]. However, their ability to compete, locate, and use habitat in the warming and drying climate of today may vary by their species-specific dispersal characteristics [17]. In general, species capable of shifting their range with changing climates can avoid extinction [18] and long distance dispersal is expected to play a critical role toward that end [19].

The spotted bat (*Euderma maculatum*) exhibits long flight capabilities and large home range sizes relative to many Vesper bats in North America [20], and has a poorly understood biogeographic legacy. Its three white, dorsal spots, large ears, and an audible, low-frequency call are charismatic hallmarks for identification [21–23]. However, this charismatic bat is rarely observed. The species occupies a patchy, widespread range, and is associated with rugged environments from central Mexico to British Columbia (BC), Canada [22, 24, 25] (Fig 1). The species largely occupies desert and xeric shrubland biome [26]. During the day, *E. maculatum* roosts in sheer cliffs [20, 21]. At night, individuals may travel over a range of elevations and vegetation types to forage, ranging from xeric lowland, subalpine meadows and pine (*Pinus*) forests to alpine meadows and spruce-fir (*Picea-Abies*) forests (-53 m to 3230 m elev.) [22, 27]. Although British Columbia, the northern-most extent of its range, is broadly situated in a temperate coniferous forest biome [26], *E. maculatum* occurs in hot dry river valleys comprised of bunchgrass, sagebrush and open canopy patches of ponderosa pine (*Pinus ponderosa*) and Douglas-fir (*Pseudotsuga menziesii*) trees [28, 29]. In southwestern North America, *E. maculatum* is capable of engaging in long, nightly flights ($\geq$ 70 km roundtrip) and uses ponds situated in open meadows [20, 27, 30]. The cryptic nature of this species has led to a limited number of specimens for study [31] and information is largely from hotspot localities where bats have a higher likelihood of capture or where bats were found dead or dying [32]. This dearth of information has led most wildlife agencies in the western United States to recognize the species as one in need of special management [33, 34]. Canada lists the species as one of special concern [35].

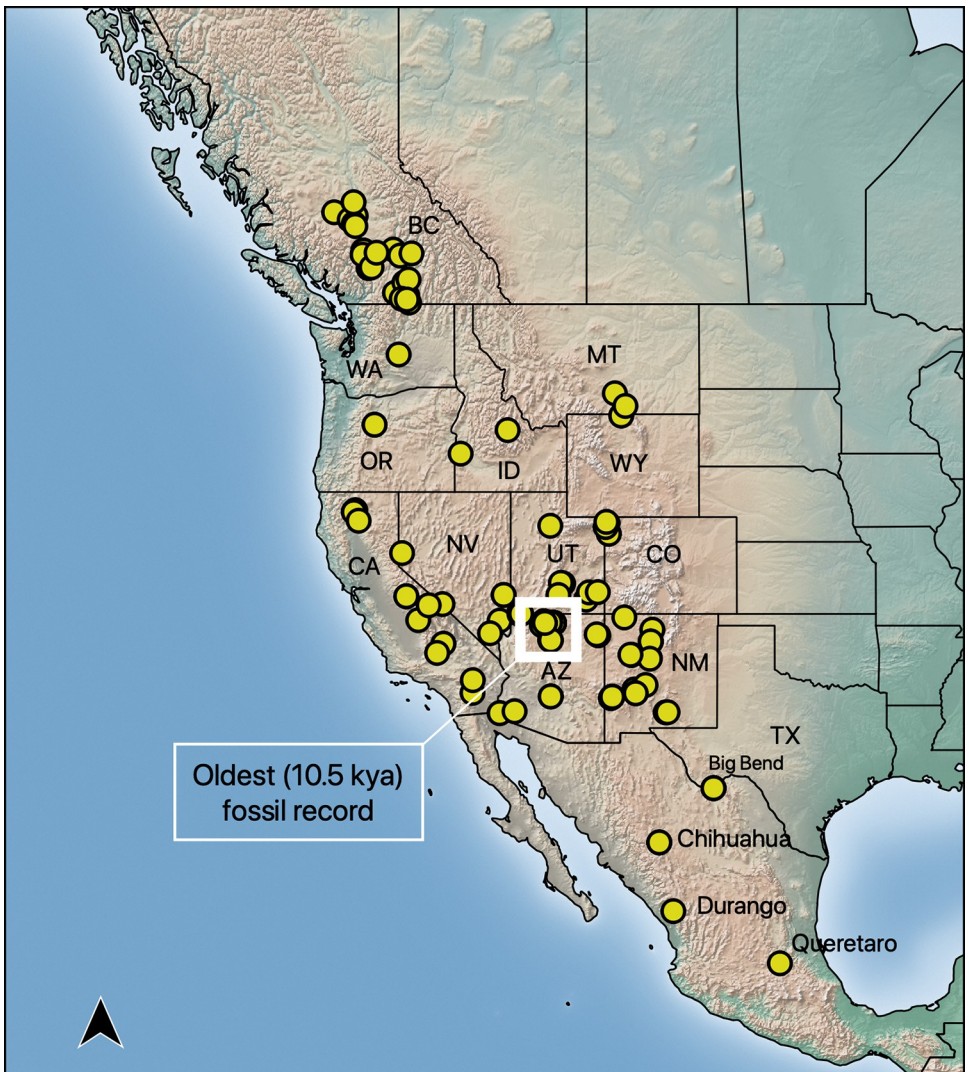

**Fig 1. Map including historic (museum specimen records), contemporary capture, and acoustic occurrences of** *Euderma maculatum*. States, provinces, or key localities where the spotted bat (*Euderma maculatum*) occurs are annotated.

The species is believed to have originated in Mexico and much of the climate of its contemporary range was likely inhospitable prior to the Holocene [36]. This is suggested by a fossil record of a mummified specimen from the southwestern United States approximately 10.5 Kya and the extant use of that locality by *E. maculatum*. An analysis of morphological traits suggested strong geographic variation [37]. When geographic groups were assumed *a priori*, the morphologies revealed that leading edge specimens of *E. maculatum* consistently formed a distinct cluster and those of the southern-most extent, including Querétaro, formed a potential rooting point. Despite the geographically informative clusters, intra-geographic structure and the phylogeographic relationships of those in the core range remain unresolved.

The complementarity of phylogeography, reconstructions of demographic history, and ecological niche modeling (ENM) can further resolve the biogeographic legacy of *E. maculatum*. In combination with phylogeography, hindcasted ENMs have revealed glacial refugia in *Rhinolophus ferrumequinum* in east Asia [38], and allowed interpretation of postglacial colonization

of *Barbasetlla barbastellus* [39] and European *Plecotus* [40]. A global analysis of bat species revealed demographic expansion of species in temperate regions and demographic bottlenecks of species in neotropical regions [41]. Such analyses could determine whether Mexico was the ancestral range for *E. maculatum* and the extent to which Holocene warming influenced its wide latitudinal range.

Our work addresses how a changing post-glacial climate might have influenced the biogeographic and demographic history of *E. maculatum*. Spanning the latitudinal range of the species, we used specimen records and their mtDNA to prime future investigations of the phylogeography and give insight into its Holocene range expansion. First, we asked whether the inferences from the *E. maculatum* fossil record could be supported by paleodistribution models (i.e., was Mexico the Pleistocene range and was habitat in the US and Canada unsuitable?). We examined this by modeling climate suitability from the last interglacial (LIG, ~130 Kya), the last glacial maximum (LGM, ~22 Kya), the mid-Holocene (MH, 6 Kya), and contemporary climate space. Second, we asked whether maternal genetic data could resolve phylogeographic structure in *E. maculatum* or if phylogenetic pattern is consistent enough for calibrating the divergence of their lineages (i.e., to estimate timing of colonization history). Third, we examined demographic history to identify any changes in historic population sizes ($N_E$) and whether they are consistent with recent transitions in climate eras. Given the latitudinal extent of the specimens, we hypothesized that mtDNA would resolve finer phylogeographic structure for leading edge (British Columbia, Canada), rear edge populations (Querétaro, Mexico), and a core range. To do this, we assessed phylogeographic structure and substitutions. Finally, we discuss how such a highly vagile bat species might track changes in one of the world's regions most impacted by climate change.

## Materials and methods

### Ecological niche modeling

The context of our ENM was based on occurrences encompassing the core range (S1 Fig in S1 File) and from foraging sites (i.e., mist-net captures). We performed all data pre-processing and spatial analyses in R v.3.6.3 [42]. We assembled geographic coordinates from 113 *E. maculatum* specimens (S1 File) based on field capture and museum collections. Specimen coordinates were derived from Vertnet (vertnet.org), Arctos (arctosDB.org), and field captures. We filtered out early records that were of *E. maculatum* found dead or dying [25, 43] in areas that may not represent elements of habitat for foraging and roosting (e.g., fringes of elevation range [44], driveways [43], and warehouses [45]). We also filtered out occurrences made vague out of concern for the animals and sites [46, 47]. We did not include acoustic occurrences because of uncertainties between where an animal was traveling and foraging. Under our target context, a single occurrence remained at one of the most northern portions of the range (Lillooet, BC, CAN). However, this occurrence was omitted because it could be due to finer-scale temporal variability at the leading edge and therefore may not represent its historic climate tolerance [48]. Under this setting, occurrences may still exhibit biases because they may represent hotspots for *E. maculatum* activity or preferred capture locations so we dereplicated occurrences within 1 km$^2$ grid cells. Finally, we retained occurrences after 1970 to match the range of years used to generate bioclimatic variables [49], resulting in 40 distinct occurrences for modelling the ENM (S1 File).

To build the ENM, we first selected all bioclim variables as potential predictors at 30 arc second resolution [49]. To avoid multi-collinearity among predictors in training the model, we extracted and Z-transformed raster values of the occurrences and conducted an analysis of principal components using the Pysch package [50]. We then selected representative

predictors from 4 principal components: mean annual temperature (bio1), mean diurnal range (bio2), isothermality (bio3), and precipitation coldest quarter (bio19). The predictors were selected toward the goal of predicting spatial pattern as opposed to interpretation of environmental parameters. We modeled a presence-only ecological niche using Maxent v3.4.6 [51] in the biomod2 package [52]. Maxent models are trained by minimizing the relative entropy between probability densities of species occurrence and environmental background in covariate space [53]. Maxent is the most widely used algorithm for prediction because it was developed to model presence-only records. The theoretical limitations and cautions of Maxent have been well documented [54, 55] but Maxent can generate useful models for rare species, particularly those with 25–50 occurrences [56]. The biomod2 package provided a platform for conducting the Maxent modeling as well as combining independent model runs for consensus projection (i.e., ensembling). We first cropped the bioclimatic predictors to obtain background points. To generate a background extent for cropping, we calculated a 1σ buffer from the means of each occurrence value [57] using altitude (alt) [49], bio2, bio3, and bio19. This allowed us to generate a background that exceeded the constraints of the occurrences to provide more flexibility in generating background points. A map illustrating the cropped background extent and geographic occurrences used for model training is available in S1 File. We trained Maxent models with an 80:20 split of the occurrences for training and testing. We used three sets of randomly sampled background points (n = 120 per set) and ran the model through 10 evaluations. Each set of evaluations for a model were then combined into a full model. We evaluated the models using the area under the receiving operating curve score for training and test sets ($AUC_{train}$, $AUC_{test}$) and the true skill statistic (TSS). For ensembling, we included any full model with $AUC_{test} > 0.7$ for projection into contemporary, mid-Holocene (~ 6 Kya, CCM4), last glacial maximum (~ 22 Kya, CCM4), and last interglacial (~130 Kya, [58]) climate space. We used the TSS as a binary threshold to provide balance between sensitivity and specificity for contrasting gained, retained, and receded climate space.

### Genetic sampling and mtDNA sequencing

We acquired 34 *E. maculatum* tissues and 1 *Idionycteris phyllotis* tissue from live capture and museum specimens that covered the entire known latitudinal range of the focal species (Table 1). *Euderma maculatum* is a monotypic genus and therefore lacks congenerics for use as a phylogenetic outgroup. *Idioynycteris phyllotis* (also a monotypic genus) is the most closely related species and is the only taxon known to be monophyletic with *E. maculatum* [59–61], having diverged approximately 20 Mya. We captured individuals between 2009 and 2014 using mist nets and collected wing biopsies and buccal swabs with the approval of the Northern Arizona University Institutional Animal Care and Use Committee (07–006, 07-006-R1, 07-006-R2; Walker et al., 2014). Four of these individuals occupied the same site where the mummy was discovered. In Canada, wing biopsies were collected under a British Columbia research permit (KA14-148262-1). We handled bats following the guidelines of the American Society of Mammalogists [62]. Animals were safely released after sample collection. Death was not an endpoint in this study. For any animals with obvious outward signs of disease, infection or injury, euthanasia was an endpoint. Isoflurane would be administered until death with cervical dislocation as assurance of death after euthanasia (method approved by the American Veterinary Medical Association; http://www.avma.org/issues/animal_welfare/euthanasia.pdf). If animals were sampled using skin biopsy punches, they were kept in a quiet, dark space with movement restrained during the procedure. Afterwards, they were treated with antibiotic ointment at the puncture site and released at point of capture when processing was completed. Museum tissues were skin clips excised from the midline of specimens originally collected

**Table 1. *Euderma maculatum* specimens used for genetic analysis and their geographic origin.** Sample IDs in italics represent specimens that were only able to be sequenced for D-loop, whereas, all others were sequenced at both D-loop and cytB. Any specimen with a holding under the Bat Ecology & Genetics Lab indicates an individual that was genetically sampled in our field surveys.

| Species | Country | Sample ID | Collection year | State/Province | Holding |
|---|---|---|---|---|---|
| *Euderma maculatum* | Canada | 140804_03_F_Canada | 2014 | British Columbia (BC) | Bat Ecology & Genetics Lab—C. Lausen |
| | | 140818_022_M_Canada | 2014 | British Columbia (BC) | Bat Ecology & Genetics Lab—C. Lausen |
| | | 140818_033_F_Canada | 2014 | British Columbia (BC) | Bat Ecology & Genetics Lab—C. Lausen |
| | | 140818_04_M_Canada | 2014 | British Columbia (BC) | Bat Ecology & Genetics Lab—C. Lausen |
| | | 140818_05_F_Canada | 2014 | British Columbia (BC) | Bat Ecology & Genetics Lab—C. Lausen |
| | | 140818_06_F_Canada | 2014 | British Columbia (BC) | Bat Ecology & Genetics Lab—C. Lausen |
| | | 140818_07_F_Canada | 2014 | British Columbia (BC) | Bat Ecology & Genetics Lab—C. Lausen |
| | United States (US) | SMNH:EUMA:TSM306333 | 2013 | Washington (WA) | Slater Museum of Natural History, University of Puget Sound |
| | | *MSU:531* | 1949 | Montana (MT) | Montana State University Vertebrate Museum |
| | | MSB:Mamm:114512 | 1990 | Wyoming (WY) | Museum of Southwestern Biology |
| | | UWBM:Mamm:82236 | 2012 | Oregon (OR) | Burke Museum, University of Washington |
| | | UWBM:Mamm:82237 | 2012 | Oregon (OR) | Burke Museum, University of Washington |
| | | MSB:Mamm:107557 | 1981 | Colorado (CO) | Museum of Southwestern Biology |
| | | MSB:Mamm:22756 | 1994 | Colorado (CO) | Museum of Southwestern Biology |
| | | MSB:Mamm:112057 | 1987 | Colorado (CO) | Museum of Southwestern Biology |
| | | MSB:Mamm:121373 | 1994 | Utah (UT) | Museum of Southwestern Biology |
| | | NMMNH:Mamm:4059 | 2000 | Nevada (NV) | New Mexico Museum of Natural History and Science |
| | | Bat15_2010_NR_M | 2010 | Arizona (AZ) | Bat Ecology & Genetics Lab |
| | | Bat16_2010_NR_M | 2010 | Arizona (AZ) | Bat Ecology & Genetics Lab |
| | | Bat4_2010_Cave_M | 2010 | Arizona (AZ) | Bat Ecology & Genetics Lab |
| | | 201_2013_Cave_F | 2013 | Arizona (AZ) | Bat Ecology & Genetics Lab |
| | | 202_2013_Cave_F | 2013 | Arizona (AZ) | Bat Ecology & Genetics Lab |
| | | 205_2013_Cave_M | 2013 | Arizona (AZ) | Bat Ecology & Genetics Lab |
| | | Bat1_2009_SR_F | 2009 | Arizona (AZ) | Bat Ecology & Genetics Lab |
| | | 200_2013_SR_F | 2013 | Arizona (AZ) | Bat Ecology & Genetics Lab |
| | | *LACM:9823* | 1953 | Arizona (AZ) | Natural History Museum of Los Angeles County |
| | | MSB:Mamm:135536 | 1994 | New Mexico (NM) | Museum of Southwestern Biology |
| | | NMMNH:Mamm:1901 | 1992 | New Mexico (NM) | New Mexico Museum of Natural History and Science |
| | | *KU:119275* | 1968 | Texas (TX) | University of Kansas Biodiversity Institute Mammalogy Collection |
| | | *MVZ:Mamm:139209* | 1931 | California (CA) | Museum of Vertebrate Zoology, UC Berkeley |
| | Mexico (MEX) | *LACM:13855* | 1961 | Chihuahua (CI) | Natural History Museum of Los Angeles County |
| | | *LACM:13856* | 1961 | Chihuahua (CI) | Natural History Museum of Los Angeles County |
| | | *TCWC:Mammals:26538* | 1972 | Querétaro (QRO) | Texas A&M University Biodiversity Research and Teaching Collections |
| | | ROM:ASK0692 | 1984 | Querétaro (QRO) | Royal Ontario Museum |
| *Idionycteris phyllotis* | | Bat2_IDPH_1a | 2009 | Arizona (AZ) | Bat Ecology & Genetics Lab |

between 1931 and 2013. We extracted genomic DNA from wing biopsies of live-captured individuals using the DNeasy Blood and Tissue protocol (Qiagen, Valencia, CA, USA). DNA from museum samples was extracted using previously described phenol-chloroform methods [63]. We amplified a 193 bp fragment of the D-loop region using custom primers (F: Dloop_196bp, `GATGCTTGGACTCAACACTG`; R: RevL16517_600, `GTCCTGTAACCATTAAGTTCAC`). PCRs contained 2 μL gDNA, 1 μL 10X Mg-free PCR buffer (Invitrogen, Thermo Fisher Scientific, Waltham, MA, USA), 2 mM MgCl$_2$, 0.2 mM of each dNTP, 0.3 U/μL Platinum Taq polymerase, and 0.5 μM of each primer in a 10 μL reaction. Thermal cycling conditions involved a 6 min denaturation at 95˚C followed by 45 cycles of denaturation at 95˚C for 30 s, annealing at 58˚C for 30 s, and polymerase extension at 72˚C for 30 s, with a final extension at 72˚C for 10 min. We also amplified a 596 bp fragment of cytochrome *b* (cytB) using custom primers (F: EUMA_Cytb_190f, `GCTCCGTAGCCCACATTTGC`; R: EUMA_Cytb_1027r, `TGGCTGTCCA ATTCAGG`). These PCRs contained 2 μL gDNA, 2.5 μL 10X Mg-free PCR buffer (Invitrogen, Thermo Fisher Scientific, Waltham, MA, USA), 3 mM MgCl2, 0.2 mM of each dNTP, 0.1 U/ μL Platinum Taq polymerase, 0.02 μg/μL of Bovine Serum Albumin, 0.5 μM of each primer in a 10 μL reaction. Cycling conditions included an initial denaturation of 95˚C for 10min, then 40 cycles of denaturation at 95˚C for 30 s, annealing at 62˚C for 30 s, extension at 72˚C for 2 min, and a final extension of 72˚C for 10 min. Amplicon was purified using the ExoSAP-IT cleanup protocol (Affymetrix, Santa Clara, CA, USA) and subsequently Sanger-sequenced using BigDye v.3.1 Terminator kit according to the recommended protocol of the manufacturer (Applied Biosystems, Foster City, CA, USA) on an ABI3130 Genetic Analyzer (Applied Biosystems, Foster City, CA, USA). We used MJ Research PTC-200 thermal cyclers for all PCRs. Sequences were edited using Sequencher v5.3 (http://www.genecodes.com). DNA sequences (D-loop, cytB) generated in this study are available in GenBank under the following accession numbers: OP234456:OP234511.

## Phylogeographic analysis

We aligned sequences of each marker using ClustalW [64] in MEGA7 [65]. Using only sequences from *E. maculatum*, we summarized genetic diversity for each marker by number of segregating and parsimony informative sites, number of haplotypes, haplotype diversity, and nucleotide diversity using R packages pegas and ips [66, 67]. We also built a parsimony haplotype network [68] using the haploNet function in pegas. We were unable to sequence five of the specimens with the cytB marker likely due to the large size of the amplicon and the ages of the specimens (collected between 1931 and 1972). Because four of these specimens were sampled from unique localities (MT (n = 1), TX (n = 1), CA (n = 1), CI (n = 2), and QRO (n = 1); an abbreviation key can be found in Table 2; geographic context can be found in Fig 1), we calculated the haplotype network using only sequences from the D-loop marker so that these specimens could be included.

Using *I. phyllotis* as an outgroup, we estimated substitution phylogenies (substitutions/site) using maximum likelihood and Bayesian methods. We concatenated both mtDNA markers

**Table 2. Polymorphism summaries among 789 sites for D-loop and cytB, used separately or in concatenation for 27–34 *Euderma maculatum* individuals (N$_{Seq}$).** This includes segregating sites (*S*), parsimony informative sites (PIS), haplotypes (Hap), haplotype diversity (Hd), and nucleotide diversity ($\pi$). We separately summarized one D-loop dataset that included more individuals, which we used to construct a haplotype network (hap).

| Marker | N$_{Seq}$ | S | PIS | Hap | Hd | $\pi$ | Sequence length |
|---|---|---|---|---|---|---|---|
| D-loop (hap) | 34 | 37 | 31 | 19 | 0.918 | 0.046 | 193 |
| D-loop | 27 | 35 | 18 | 15 | 0.889 | 0.043 | |
| cytB | 27 | 22 | 5 | 13 | 0.852 | 0.005 | 596 |

(total positions = 776 after removal of trailing indels between *E. maculatum* and *I. phyllotis* in the D-loop alignment) with substitution models estimated for each marker using BIC-based model selection in jModelTest v.2.1.10 [69, 70]. Based on those results, we estimated phylogenies using the K80+*G* substitution model for D-loop and TPM2uf+*G* for cytB. To calculate a maximum likelihood (ML) tree, we used RAxML-NG v.1.1 [71] with 1000 bootstrap iterations. For a Bayesian tree, we used MrBayes v.3.2.7 [72] in two runs for 12,000,000 generations and sampled every 1000. Chains for each run were inspected for convergence using Tracer v1.7.1 [73]. If convergence was reached, we estimated a strict consensus tree, omitting the first 30% of sampled trees as burn-in. For an additional Bayesian substitution tree, we used BEAST2 v.2.7.3 [74]. Clock and tree models were linked whereas site models (i.e., substitution models) were unlinked to parameterize individual substitution models and estimate relative substitution rates between both mtDNA markers. We used an optimized relaxed clock and specified that the clock rate be estimated. We initially compared three coalescent tree priors, assuming either a constant population, exponential population, or Bayesian skyline using default parameters. We compared these models using path sampling, calculating Bayes factors from marginal likelihood estimates between model pairs. Based on those results, we ran the analysis in four independent, differently seeded runs, including an additional run that only sampled from the prior. To scale the branch lengths by substitutions/site, we set 'substitutions ="true"' in the TreeWithMetaDataLogger section of the XML file. We ran for 30,000,000 chain iterations and sampled every 1000. Upon inspecting for convergence using Tracer v1.7.1 [73] and effective sample sizes (ESS) > 200 for parameter estimates, trees among runs were combined to estimate a maximum clade credibility tree (first 30% of burn-in was removed). We used R package phytools [75] to plot geophylogenies with outgroups pruned from the tree for clarity. We also used FigTree v1.4.3 [76] to visualize phylogenies with the outgroup (S2 File). Topologies were assessed based on discretized range groupings (Northern, Central, and Southern), which were inferred from haplotype structure, consistent clades among the phylogenies, and geographic context.

## Demographic history

We reconstructed historical demography using an extended coalescent Bayesian Skyline analysis [77] in BEAST2 v2.5.2 [74]. Using both mtDNA partitions, we only included *E. maculatum* (n = = 27) in the analysis. Because we modified the sample set, we re-estimated substitution models using jModelTest v.2.1.10 [69, 70]. Based on these results we used the K80+*G* substitution model for D-loop and the TPM3uf model for cytB. Site and Clock models were unlinked, whereas tree models were linked. We used a strict clock model, assuming a clock rate of 3.5% per million years for cytB [78], whereas the rate for D-loop was to be estimated relative to cytB. This clock rate was based on the cytB sequences of a related genus, *Plecotus* (common ancestor shared ~26 Mya [60, 79, 80]), which to our knowledge is the most reasonable clock rate for *E. maculatum* given a lack of such information for the focal species or those more closely related. We set the population factor for both mtDNA markers to 0.5 to specify haploid, maternal inheritance. To improve mixing success, the populationMean.alltrees parameter was given a normal distribution with a mean of 1.0 and standard deviation of 0.1. Also, to improve convergence, we specified relative substitution rates (D-loop = 2.878; cytB = 0.433) based on estimates from BEAST2 runs described above. All other parameters and operator values remained at default settings. We ran the chain, sampling every 1000 iterations until convergence was reached (25,574,000 iterations) and ESS for parameter estimates were > 200. Following the run, we calculated 95% central posterior density intervals using functions from plotEBSP.R (https://evomics.org/). We then scaled the population size parameter (θ) to reflect effective

population size ($N_E$), assuming an average generation time of 2 years. We used R package ggplot2 [81] for visualizing the Skyline plot. In addition to the Skyline plot we also evaluated demographic history with neutrality tests in DNASP v.6.12 [82]. We tested with Tajima's $D$ [83] and Fu's $F_S$ [84]. Because these tests are sensitive to synonymous and non-synonymous mutations, we only used cytB as a coding region. Under a null hypothesis of static demography, we tested significance of observed estimates of θ (alpha = 0.05) with 10,000 coalescent simulations using the 1-locus, 1-population model.

## Results

### Ecological niche model

ENMs generated from two of three datasets of background points gave test AUC scores (Table 1 in S1 File) of 0.81 ($AUC_{train}$ = 0.81) and 0.83 ($AUC_{train}$ = 0.79), indicative of informative models [85]. The similar magnitudes of $AUC_{test}$ and $AUC_{train}$ of these models indicated that the models were not overfit. The model that we omitted from the ensemble had an $AUC_{test}$ of 0.56 and $AUC_{train}$ of 0.83, which indicated that this model was overfit and predicted poorly. Each predictor substantially contributed to the models. The average importance of the predictors was isothermality (0.60 ± 0.11 SD), mean diurnal temperature (0.49 ± 0.35), mean annual temperature (0.31 ± 0.09), and precipitation coldest quarter (0.18 ± 0.1) (Table 2 in S1 File). Despite some variability among model performance, continuous distribution maps (S1 File) encompassed an area largely occupied by *E. maculatum* and included known locations that were omitted in the pre-processing stage.

It is preferable to present a Maxent map as a continuous gradient of relative suitability [55] (Available in S1 File) but we set a binary suitability threshold to better summarize general patterns of gained, receded, and retained climate space (Fig 2). The TSS was between 0.48 and 0.54 so we set the binary threshold at 0.5 to reflect a balance between sensitivity and specificity for plotting. The comparison between the LIG and LGM (~130 to 22 Kya) showed that historic

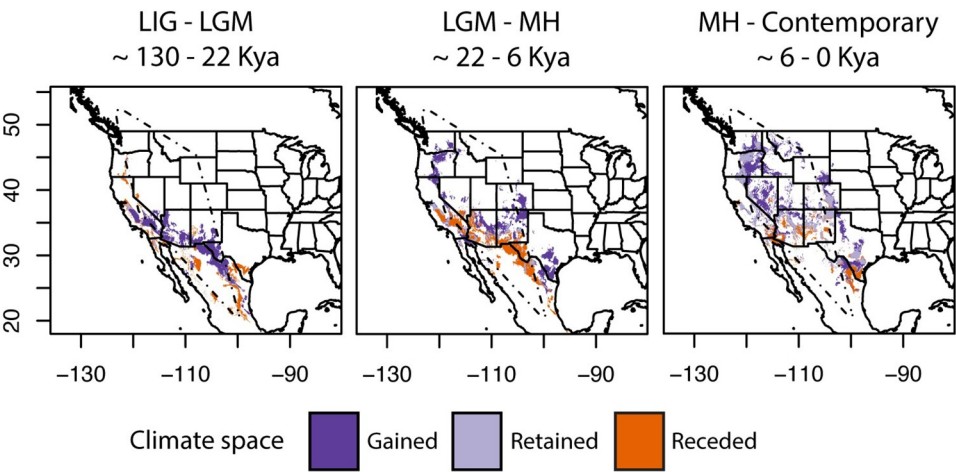

**Fig 2. Gained, retained, and receded climate space for *Euderma maculatum*, predicted from ecological niche modeling (Maxent) (binary threshold = 0.5).** Gained climate space is in dark purple, retained climate space in light purple, and receded climate space in orange. Left to right panels: a stretch of receded climate space spans from the general proximity of central Mexican specimen, approximately along the Sierra Madre Oriental to the lower southwestern United States at the transition of the last interglacial (LIG) and last glacial maximum (LGM), receding by the mid-Holocene (MH) and into contemporary climate space. The majority of gained climate space into the contemporary range can be observed by the mid-Holocene and within the last 6000 years. A convex polygon in dashed lines border the extent of the known species range.

climate space was largely suitable in Mexico, indicating a larger expansion of suitable climate into the southwestern US at the height of the Wisconsin glaciation. By this time, climate space began receding from the most southern extent of the species range (Querétaro, Mexico). A comparison of projections between the LGM and the mid-Holocene (~22 to 6 kya) showed a more substantial expansion and contraction of predicted climate space. Climate space receded from central Mexico into the southwestern US, whereas climate space expanded from the southwestern US in a narrow strip along the US States of CA and NV into OR and WA. A comparison of projections between the mid-Holocene and contemporary climate space (~6 to 0 kya) showed inward, patchy expansion across western North America. Patterns of receding climate space occurred in the southern range with some gained climate space approaching the Canadian range. The contemporary projection largely predicted suitable climate space within the boundaries of the known range with some under-prediction into the leading edge range of Canada and some potential over-prediction outside the eastern margins of the known *E. maculatum* range.

## Phylogeography

We recovered 19 D-loop haplotypes (193 positions) with 57 segregating sites and 23 parsimony informative sites when both D-loop and cytB were considered (Table 2). Prominent clusters in the parsimony network (Fig 3) were unique to at least three geographic regions. Within at least three mutational steps, these clusters included a northern haplogroup (BC, WA, OR); a genetically, regionally diverse, and star-like central range haplogroup (northern Mexico, southwestern US and other ranges along the Rocky Mountains, US); and a single haplotype in central Mexico, shared by both Querétaro specimens. This southern range haplotype was the most distinct with 17 mutational steps from the nearest haplotype (CA). Surprisingly, northern Mexican (CI) haplotype (n = 2) had much fewer mutational steps to the central range haplogroup than to the Querétaro haplogroup. Other notable haplotypes included those sampled from CA, TX, WY, MT, and CO, although each was represented by only a single specimen.

Three substitution phylogenies (Fig 4) based on maximum likelihood and Bayesian optimality criteria supported the three regional groupings of the haplotype network but gave different basal relationships when rooted to the outgroup. RAxML-NG produced a topology that was rooted to a specimen from CO, BEAST2 produced a topology that rooted at the Querétaro specimen, and MrBayes gave a toplogy that was more generally rooted at a polytomic region containing specimens from the central range haplogroup. The observed incongruency could be due to multiple factors including lack of more conservative nuclear markers, long-branch attraction in Bayesian trees, the relatively distant relationship to the outgroup, non-bifurcating patterns of divergence, or an incomplete specimen record. Although we were unable to resolve deeper phylogenetic relationships using mtDNA, we were able to observe some phylogeographic consistencies. Each tree exhibited a clade monophyletic to the northern-most portion of the *E. maculatum* range. Specimens from the northern-most portion of the range formed a monophyletic group (BC, WA, OR) with 84% bootstrap support and 77–100% posterior probabilities for this clade, suggesting that this could be considered a leading edge clade. As with the haplotype network, the Querétaro lineage exhibited a markedly longer branch length than others. However, the unresolved relationship confounded interpretation of colonization history and prevented divergence dating.

## Historical demography

The extended Bayesian Skyline analysis supported a Holocene-era demographic expansion (Fig 5). Effective population size began increasing ~26 Kya at the onset of the last glacial

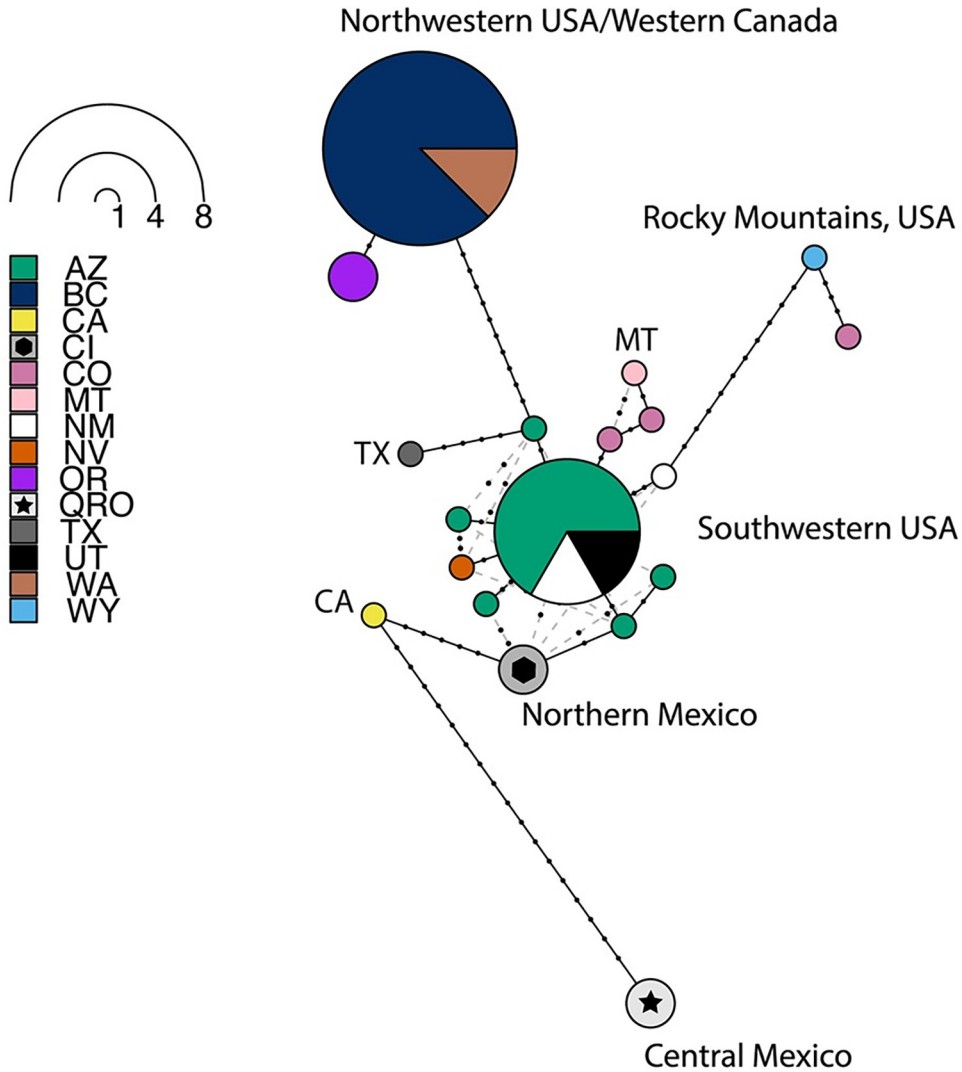

**Fig 3. Maximum parsimony network for 34 *Euderma maculatum* individuals using only the D-loop marker.** Pies and slices are colored by 14/16 states/provinces the species is in known to occur.

maximum. This indicated that demographic expansion occurred during the Holocene. As per the ENM, this expansion coincided with a consistent increase in suitable climate space into contemporary periods. Although CPD intervals of the Skyline plot were relatively wide, expansion was supported by both neutrality tests (Table 3). This was indicated by the negative signs of the test statistics as well as a rejection of the null hypothesis of static demography.

## Discussion

Our work provides a window into understanding climate-induced range expansion and contraction for a long-distance flying bat species in the absence of contemporary, anthropogenic pressures. Our results support the hypothesis of Mead & Mikesic [36] that Mexico was to a large extent the Pleistocene-era range of *E. maculatum*, with the species then expanding into most of its US and Canadian range during the Holocene. In opposition to our second hypothesis, a lack of phylogenetic certainty confounds a more precise interpretation of the colonization

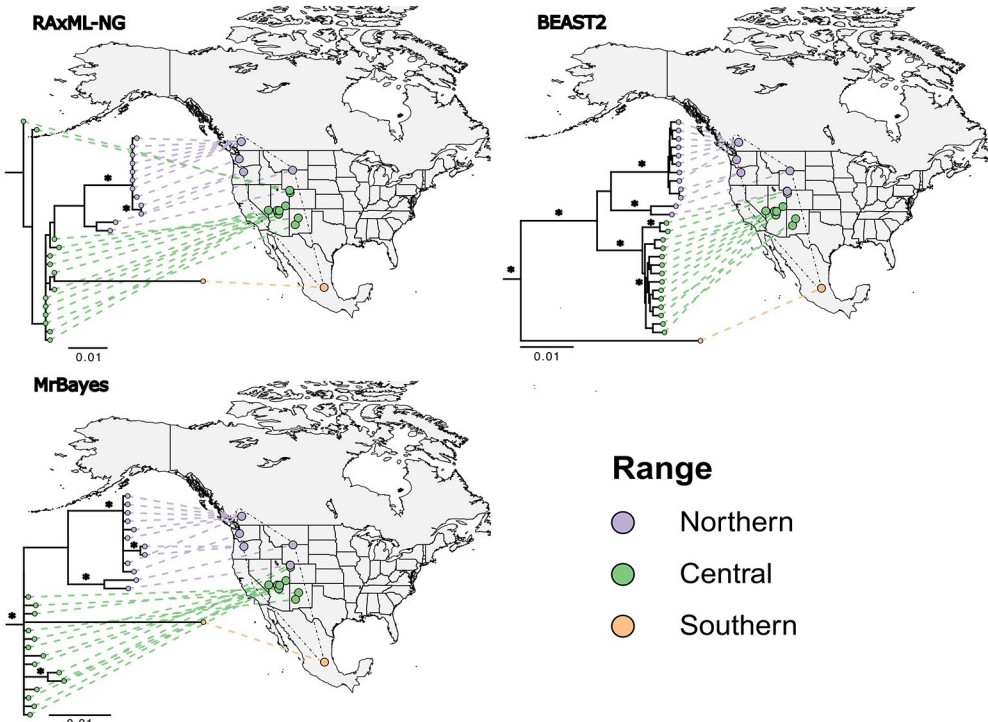

**Fig 4. Incongruent phylogenetic trees (branch lengths = substitutions/sites) for 27 *Euderma maculatum* individuals (D-loop and cytB) estimated from maximimum likelihood (RAxML-NG) and Bayesian methods (MrBayes and BEAST2).** Geophylogeny vectors from each tip to its corresponding geographic origin are colored by discretized range categories (Northern range = purple, Central range = green, and Southern range = orange). An asterisk indicates ≥ 70% bootstrap clade support for maximum likelihood and ≥ 80% for clades supported by Bayesian posterior probabilities.

history. Still, broader haplotype patterns suggest a stable rear edge lineage in central Mexico, a diverse and polytomic core range largely throughout the southwestern US, and a leading edge lineage into British Columbia.

The idea that Mexico was the Pleistocene-era range of *E. maculatum* was originally founded on the occurrence of an early Holocene-era mummy and absence of Pleistocene-era occurrences from hotspot localities (e.g., caves and middens), ranging from the Grand Canyon to southeastern Arizona [36, 87]. The ENM allowed us to predict that contemporary climate space largely encompasses the mountainous periphery of the broader Great Basin (Fig 2). Nevertheless, hindcasting into the LIG and LGM indicates that climate space mostly encompassed the mountainous regions of the Mojave and Chihuahua desert regions, mainly along the Sierra Madre Occidental. We were able to predict climate space into the Querétaro locality (the southern-most margin of the range) despite training the ENM without the Querétaro occurrence, which lends strength to the generalizability of the ENM. Phylogeographic evidence shows that the Querétaro haplotype exhibits a high amount of substitutions per site, even relative to specimens from northern Mexico. The northern Mexican (CI) haplotype was more similar (within 2–3 mutational steps) to those commonly occurring in the southwestern US. Evidence from the ENM and mtDNA suggests that the Querétaro lineage may have experienced isolation upon contraction of suitable climate, beginning at least 130 Kya. Otherwise, the distinctiveness of this lineage as well as the uncertainty of its relationship to other haplotypes could reflect incomplete sampling in Mexico. This portion of the range lacks specimens, known only to occur in Querétaro, Chihuahua, Cuatro Ciénegas Basin [88], and two localities

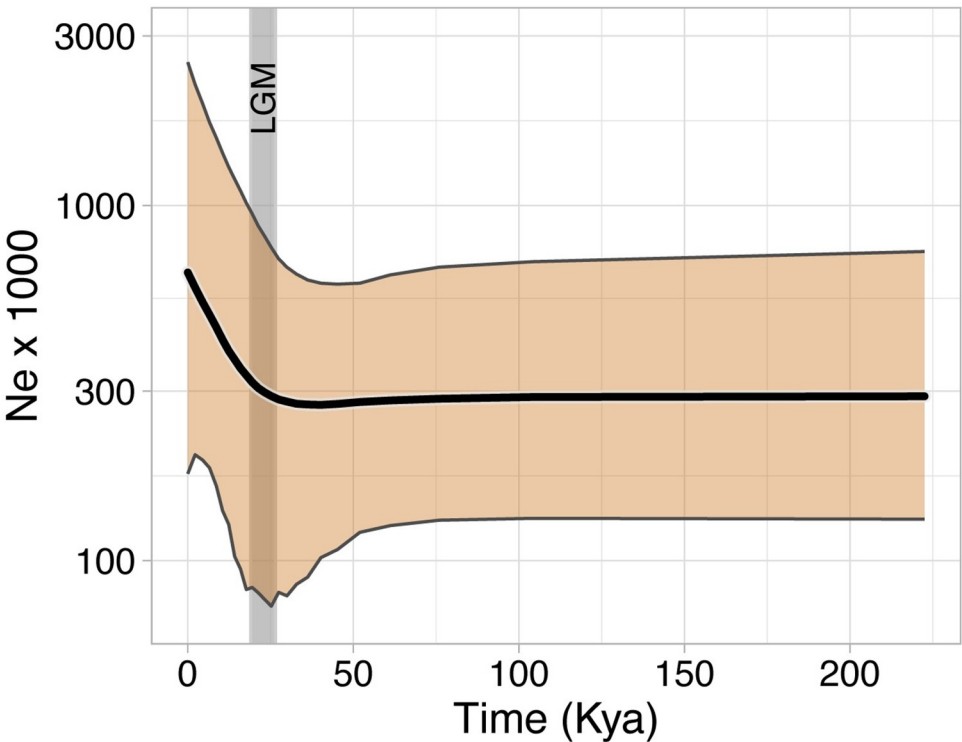

**Fig 5. Extended Bayesian Skyline plot based on partitioned D-loop and cytB sequences from 27 *Euderma maculatum* individuals.** The x-axis is in thousands of years and the y-axis (log10-scaled) is effective maternal population size scaled by an assumed generation time of 2 years. The thick black line is median $N_E$ and the orange ribbon indicate 95% central posterior density intervals. The plot is annotated with a vertical ribbon showing that population expansion began approximately by the known temporal range of the last glacial maximum, occurring between 19 and 26.5 Kya [86].

in Durango [89, 90]. Cuatro Ciénegas Basin is a well-known glacial refugium [91] immediately southeast of the hotspot locality of Big Bend, TX [22]. We were unable to acquire specimens from the Durango localities and it is possible that they can improve phylogeographic context. Occurrence in Durango was first recorded in 1965 in the Ocampo, Presa de Ojito municipality and later in La Michilía Biosphere Reserve in 1984 [92]. The latter locality would have met the criteria for inclusion in our ENM model but was absent from our initial Vertnet search and so it was omitted from model training. Although this omission could have introduced bias to the ENM, we doubt that the absence of this single point substantially influenced the broader patterns. As the most intermediate locality to Querétaro, specimens from Durango could improve phylogeographic context. Overall, the ENM indicates that Mexico provided more suitable climate space and possibly indicates that a more northerly population could have already been staged for expansion into the contemporary range. Contraction of climate space throughout

**Table 3. Neutrality test results for 27 *Euderma maculatum* individuals using cytB (596 bp).** Significance testing was based on 10,000 replicate coalescent simulations (alpha = 0.05).

| | Observed | Coalescent simulations | |
|---|---|---|---|
| Test statistic | θ | θ (95% CI) | P-value |
| Tajima's $D$ | -1.881 | -1.734 to 1.884 | 0.0141 |
| Fu's $F_S$ | -5.383 | -4.051 to 5.061 | 0.005 |

the Holocene could have led to genetic isolation in Querétaro, at least in terms of maternal inheritance.

The majority of the contemporary distribution was likely an outcome of Holocene warming. This is indicated by a marked increase in climate suitability into the central and northern ranges during this period (Fig 2), along with coincident demographic expansion (Fig 5, Table 3). The star-like pattern in the haplotype network (Fig 3) is further indication of population expansion [93]. This star-like pattern is most notably centered in the southwestern US and is distributed around the highest frequency haplotype. The high frequency of this haplotype suggests that observable signals of demographic expansion likely originated from an ancestral population centered in AZ, UT, and NM by the Pleistocene-Holocene transition [94]. This is the approximate region where the 10,500 year old mummy was discovered (Fig 1) and an expansion of climate space would allow greater access to the rich cliff habitat of the US Intermountain West. Without the ability to date divergence (given incongruent phylogenetic topologies), it is challenging to precisely determine when the leading edge range was colonized. The absence of suitable climate space in BC may suggest relatively recent colonization. The ENM underpredicted range into BC but we believe this could be because species on the leading edge might persist in fine-scale, temporally variable, microclimate conditions [48]. The lack of prediction space at the leading edge could also be explained by only having trained the ENM using central range occurrences. However, this underprediction was minimal given the close proximity of predicted suitable climate space to known range in BC. Subsequent Holocene warming would have likely allowed the species to better expand its range inwardly throughout the Great Basin, particularly within the last six thousand years. Taken together, our results support interpretations from the fossil record that contemporary biogeography is a product of Holocene warming [36] but that the southwestern US can be considered core range from which geographic expansion occurred.

*E. maculatum* likely owes its response to climate-induced change in part to its exceptional flight capabilities; however, its powered flight may be both the exception and the rule. We emphasize that our study represents an example of the historic range expansion of a western North American bat with great dispersal capabilities. Persistence throughout its wide range has also been explained by its varied diet of noctuid, geometrid, and lasiocampid moths [95]. Still, previous phylogeographic studies of North American desert bats have largely focused on species with relatively low vagility and were largely based on interspecific divergence throughout the Pleistocene [96–98]. It is still unclear if or how the dispersal ability of *E. maculatum* contributed to the rate of poleward expansion throughout the Holocene.

The phenomenon and rate of range expansion depends on many factors in addition to dispersal capability. Leading edge populations, like populations of *E. maculatum* in BC, are primarily expected to influence the rate of future range shifts [99]. Low numbers of reproductive individuals with or without sex-bias may slow the rate of expansion and these individuals may benefit from reduced resource competition [100]. There is a key contrast of dispersal between the leading edge and core range populations of *E. maculatum*. Individuals in the core range appear to use their flight for travel to foraging areas, whereas those at the leading edge maximize their flight for foraging locally. In the high elevation semi-deserts of the core population, individuals regularly returned to a known foraging site, traveling straight-line distances of 77 km (round trip), and had an average home range of roughly 300 km$^2$ [20, 30]. In the leading edge (Fraser and Okanagan valleys, BC), individuals also tended to return to the same local foraging sites [23, 101] but were observed roosting much closer to their foraging sites (6–10 km) [101, 102]. Although less distance is traveled in the leading edge range, the long flight is instead leveraged for foraging in continuous flight. Here, *E. maculatum* foraged over a small area, with home ranges estimated at 1.57 km$^2$–4.58 km$^2$ [103]; using radiotracking, maximum

straight-line distances covered by foraging bats ranged from 6.9 to 18.8 km, with a maximum linear distance covered during a 30 minute foraging time of less than 3 km. The smaller home ranges on the leading edge could be a result of higher prey and water densities, lower interspecific competition, or shorter nights in northern areas limiting foraging times [104]. Dispersal-promoting behaviors of individuals on the leading edge requires further investigation. An important aspect of predicting the future response of this species, with respect to range expansion, is by further comparing dispersal characteristics, traits, and behavioral tradeoffs of leading and core range lineages. Dispersal-promoting morphologies can potentially be discovered given the high variability of quantitative traits among *E. maculatum* [37].

Given the phylogeographic structure and differences in flight behaviors between core and leading edge ranges, we hypothesize that *E. maculatum* might engage in dispersal-promoting behavior in response to lower habitat quality [99] (e.g., drought, higher temperatures). The larger home ranges of those in the core range could be a result of poorer habitat quality in a drier, semi-desert southwest, whereby increased travel could reflect lower selectivity in foraging site or prey selection [105, 106] than individuals in the leading edge. Increasing temperature can lead to roost abandonment in female bats [107] and physiological responses to heat have been documented in the core range of *E. maculatum* at 30˚C [21]. We surmise, however, that canyon walls provide a cool enough respite for hot daytime temperatures and lack of water and prey may affect movements to a greater degree. Drought in deserts makes lactating female bats thirstier, expending more energy to drink [108] and can lower reproductive output [109]. This reduction in habitat quality could increase inter and intra-specific competition at foraging sites, decrease vegetation for insect prey, decrease water availability, and therefore promote increased rates of founding new localities. During a drought in 2006, *E. maculatum* was detected at 6 new foraging localities in New Mexico (core range) and was thought to be a phenomenon of the individuals flying further distances to drink [32]. Although *E. maculatum* might be able to respond to changing climates, a key limiting factor is the predominant selection of fixed roost structures (e.g., sheer cliffs, crevices), which will remain despite changing climate and vegetation [110].

Although the geographic structure of *E. maculatum* can be further refined by nuclear DNA, the lineages of the rear, leading edge, and core range provide a first step for considering population-level conservation implications. At the rear edge, the southern population (Querétaro) may meet the criteria of a climate relict [111], stable rear edge population and warrants further survey efforts. Populations of the southern lineage may be geographically isolated given recession of suitable climate space from the region in the last 130 thousand years and a high number of mutational steps to northern Mexico. The entire range of the species may be functionally misleading due to the lack of suitable climate space into central Mexico and in the patchiness throughout Chihuahua and Big Bend regions. Additionally, we are unaware of any detections in Querétaro since 1984 [46]. This population of the southern haplotype could be considered an evolutionary significant unit (ESU). The definition of ESUs have varied over the years [112] but we believe that the degree of morphological clustering [37] and a potential pattern of genetic isolation, geographic isolation, differences in climate suitability, and importance of the lineage to the evolutionary legacy of the species as outlined in our study, warrants consideration. The lineage could be important for understanding the resilience of this species as well as a possible trajectory of trailing edge populations in the northerly ranges. It could help address what might occur when climate shifts from an area but a population remains [111]. The distinctiveness of the leading edge lineage is more likely a phenomenon of drift from occasional founding events from the core range [113]. The importance of the leading edge range is for continued and future exchange of locally adapted genes (e.g., cold tolerance and dispersal-promoting) as migrants serially colonize from the core [10]. Similarly, the core range could serve

as a reservoir for warmer adapted genes with the potential to spread into the leading edge range [114]. Our results suggest that prior to the Anthropocene, *E. maculatum* has been well-suited to shifting its range to follow geographic changes in climate, a key quality towards avoiding climate driven extinction [18]. But the uncertainty of anthropogenic pressures in more localized, drought-prone, regions could still threaten the persistence of local populations [20]. In particular, over-grazing could reduce vegetation that supports their prey base of moths [95]. However, on the northern end of the range, in British Columbia, wildfires are of increasing frequency and expected to result in forest conversion to dry grassland ecosystems, especially at lower elevations in areas of rocky mountainous terrain (Utzig 2012 [Unpublished]). This suggests that suitable habitat for *E. maculatum* in British Columbia and other areas of the Pacific Northwest [115] may increase with climate change in areas where suitable rock/cliff features exist. Much of the finer scale, contemporary, genetic processes of range expansion for the species are still unknown. However, an accumulation of localized extirpations in the core range could potentially slow the rate of beneficial alleles spreading toward the leading edge [100, 113, 116].

## Supporting information

**S1 File. Maxent geographic occurrences, continuous projection maps, performance metrics, and predictor importance.**
(PDF)

**S2 File. Phylogenetic trees for *Euderma maculatum* shown in the context of the outgroup (*Idionycteris phyllotis).***
(PDF)

## Acknowledgments

We thank the New Mexico Museum of Natural History and Science, Museum of Southwestern Biology, Burke Museum, University of Washington, Montana State University Vertebrate Museum, Slater Museum of Natural History, University of Puget Sound, University of Kansas Biodiversity Institute, Museum of Vertebrate Zoology, Natural History Museum of Los Angeles County, Texas A&M University Biodiversity Research and Teaching Collections, and Royal Ontario Museum for historical tissue loans. We thank Arizona Game and Fish, USDA Forest Service, US Bureau of Land Management, US National Park Service, and the Navajo Nation for field capture permits. Thanks to Brad Butterfield for instruction on ecological niche modeling and to Crystal Hepp for advice on running BEAST2 software. We also thank two anonymous reviewers for advice on improving earlier manuscripts. This material is based upon work supported by the National Science Foundation Graduate Research Fellowship under Grant No. (NSF) 1938054.

## Author Contributions

**Conceptualization:** Daniel Enrique Sanchez, Faith M. Walker.

**Data curation:** Faith M. Walker, Cori Lausen, Carol L. Chambers.

**Formal analysis:** Daniel Enrique Sanchez.

**Investigation:** Daniel Enrique Sanchez, Colin J. Sobek.

**Methodology:** Daniel Enrique Sanchez.

**Project administration:** Faith M. Walker, Cori Lausen.

**Resources:** Faith M. Walker, Cori Lausen, Carol L. Chambers.

**Supervision:** Faith M. Walker, Carol L. Chambers.

**Visualization:** Daniel Enrique Sanchez.

**Writing – original draft:** Daniel Enrique Sanchez.

**Writing – review & editing:** Daniel Enrique Sanchez, Faith M. Walker, Colin J. Sobek, Cori Lausen, Carol L. Chambers.

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
