## [Decision Letter · Decision Letter 0]

28 Nov 2022

PONE-D-22-23686Once upon a time in Mexico: Holocene phylogeography of the spotted bat (*Euderma maculatum*)PLOS ONE

Dear Dr. Sanchez,

Thank you for submitting your manuscript to PLOS ONE. After careful consideration, we feel that it has merit but does not fully meet PLOS ONE’s publication criteria as it currently stands. Therefore, we invite you to submit a revised version of the manuscript that addresses the points raised during the review process.

We look forward to receiving your revised manuscript.

Kind regards,

Bi-Song Yue, Ph.D

Academic Editor

PLOS ONE

Journal Requirements: 

2. To comply with PLOS ONE submissions requirements, in your Methods section, please provide additional information regarding the experiments involving animals and ensure you have included details on (1) methods of sacrifice, (2) methods of anesthesia and/or analgesia, and (3) efforts to alleviate suffering

3. We note that Figure(s) 1, 2, 3, and 4 in your submission contain [map/satellite] images which may be copyrighted. All PLOS content is published under the Creative Commons Attribution License (CC BY 4.0), which means that the manuscript, images, and Supporting Information files will be freely available online, and any third party is permitted to access, download, copy, distribute, and use these materials in any way, even commercially, with proper attribution. For these reasons, we cannot publish previously copyrighted maps or satellite images created using proprietary data, such as Google software (Google Maps, Street View, and Earth). For more information, see our copyright guidelines: http://journals.plos.org/plosone/s/licenses-and-copyright.

1. You may seek permission from the original copyright holder of Figure(s) 1, 2, 3, and 4 to publish the content specifically under the CC BY 4.0 license.  

Natural Earth (public domain): " ext-link-type="uri" xlink:type="simple">http://www.naturalearthdata.com/"

3. We note that you have referenced (ie. Bewick et al. [5]) which has currently not yet been accepted for publication. Please remove this from your References and amend this to state in the body of your manuscript: (ie “Bewick et al. [Unpublished]”) as detailed online in our guide for authors

Reviewers' comments:

Reviewer's Responses to Questions

**Comments to the Author**

1. Is the manuscript technically sound, and do the data support the conclusions?

Reviewer #1: Partly

Reviewer #2: Partly

2. Has the statistical analysis been performed appropriately and rigorously? 

Reviewer #1: Yes

Reviewer #2: Yes

3. Have the authors made all data underlying the findings in their manuscript fully available?

Reviewer #1: Yes

Reviewer #2: No

4. Is the manuscript presented in an intelligible fashion and written in standard English?

Reviewer #1: Yes

Reviewer #2: Yes

5. Review Comments to the Author

Reviewer #1: Review of the article titled “Once upon a time in Mexico: Holocene phylogeography of the spotted bat (Euderma maculatum)” (ID: PONE-D-22-23686). Sanchez et al. present a survey on the biogeography of rare bat species in western North America. The idea of this study is not innovative but it is an interesting and important contribution to the knowledge of past and current distribution and diversity of Nearctic bats, particularly species of xeric environments.

This study is also helpful for proper planning of conservation for the threatened bat species as genetic data show that its protection needs to consider the spatio-genetic structure of populations and niche modeling indicate which areas are crucial for preserving the distribution of this species. However, there are also some problems with this study and some of them are important, therefore my opinion about the publication of this research is ambiguous.

First, as this study is broadly focused on niche modeling and phylogeographic data are additional I would suggest changing the title to “… biogeography … “ or informing that the study is about Holocene phylogeography and past and current niche modeling.

Next, niche modeling is explained (both in methods and results) with many details and I do not see any problems with the use of this method (with the minor question below). I would only ask why the Authors did not include in modeling also other periods – especially prediction of distribution in the future (there are available predictions of climate for at least 2050-2070), what could bring interesting information for the protection of this species.

In l. 149-150 it is informed that modeling was based on climatic data, whereas in l.160 also altitude was added to variables. It is not justified in the paper, however, for me, it is a good decision as this bat species is present mostly in mountainous areas.

Use of Idionycteris phyllotis is not justified in methods until the end (l. 247) – it should be explained in l. 171 that this species is used as an outgroup. I wonder why this species which belongs to a different genus than Euderma?

I wonder why the Authors used only a single genetic marker for phylogeography? CO1 is widely used as a barcode and marker in phylogeography, however, its utility is limited because of the mode of its inheritance. Moreover, single locus analyses could not solve the whole history and contemporary diversity of the species. It is not justified in the text why not to use other markers like some introns, microsatellites or genomic data. Especially SNPs based on NGS are now standard in any phylogenetic or phylogeographic studies. In my opinion, the use of only mtDNA is outdated and should no be longer accepted in research, maybe just in some preliminary works.

Next, for the only Mexican site available only a single individual, moreover old museum sample. I wonder if the Authors managed to sequence this sample without problems and if the reads were of high quality to be sure that all polymorphisms are real, especially since this was only one sample so there was no reference. I am pretty sure that these individuals belong to different lineages but it is not appropriate to draw conclusions from a single sample.

Regarding analyses. This is mostly phylogeographic study, whereas methods in use are more phylogenetic (tree reconstruction, dating). I wonder why the Authors did not use e.g. networks, some statistics describing current diversity (AMOVA, IBD etc.) or gene flow (simply Fst or some coalescent methods) or demography (Mismatch Distribution or skyline plots). Moreover, as 2 (or 3) distinct units were identified it would be interesting to test if they should be considered as distinct taxa, I mean especially Mexican bat (but again – it is only a single sample). About dating, I am not sure what was the reason to assume that the mummy represents the moment of split between central and northern lineages. Why its age could not be associated with earlier periods (before this split) or later periods (being members of either the central or northern clade)? Are there any morphological features of this mummy which suggest the assignment or it was based on some other data (paleontological?)?

Moreover, it is possible that in Mexico exist other lineages of this bat and more samples from various localities are needed to fully understand its phylogenetic diversity. Some of the conclusions of this study are uncertain due to that tiny sampling e.g. l. 360 about paraphyletic of S lineage – it could not be confirmed without more samples as other samples from northern Mexico could be intermediate. Generally, I am not sure if paraphyletic is an appropriate term here as all 3 lineages form a monophyletic group and members of the central and northern lineages belong to single clades so are also monophyletic. Southern lineage is just one sequence so its phylogenetic relationship to other lineages could change with more data. I am not sure also in Mexico could be assigned as an ancestral range as members of other lineages are absent in Mexico (at least based on available data), so their ancestral areas could be rather in SW USA.

Reviewer #2: Sánchez et al. conducted ENM and phylogenetic analyses to assess how post-glacial climate changes influenced the phylogeographic structure of Euderma maculatum. The manuscript is well written and interesting. There are some points that the authors should address and clarify.

In introduction, please clarify in “we asked whether maternal genetic data could better resolve phylogeographic structure”. This sentence suggests that there were previous unsuccessful attempts to resolve the phylogeographic structure of this species. Please clarify, and if there was a previous study provide background information about it.

In methods section please mention the extension to which the bioclimatic layers were cropped for ENM analyses. Also, mention the AOGCM used for past projections, and justify why not evaluating range dynamics to the LIG. In addition, it is not clear which database was searched to obtain occurrence points.

In the introduction the authors mention that E. maculatum originated in Mexico; nevertheless, only one sample from Mexico was included in the study. In a quick search in GBIF I notice that there are additional occurrence points from Mexico, was this database searched or were these points excluded for some reason? Please, in methods section, justify or explain why only one point from Mexico was used to obtain ENM. Moreover, in the discussion section, mention the limitations of lacking samples from the center of origin (where ancestral haplotypes are expected) in the ENM and phylogeographic analyses. Particularly as a lack of information is not the same as a true absence, and this cause bias in ENM models and past projections.

Apparently, there is some correspondence between the northern and central lineages and habitat type (temperate forest and xeric shrubland; but not sure as habitat type is not depicted in Fig. 3); If so, it would be interesting to test for niche differentiation between lineages.

Also, please provide a haplotype network, as this type of analysis depicts genealogical relationships among haplotypes and also could provide some clues regarding the historical demography of this lineages, as we could expect to find signals of population expansion that are consistent with the range expansion mentioned in the discussion.

For the organization of the manuscript, I suggest moving the corroboration of the calibration assumptions before presenting the final phylogeny as this analysis provide support to the phylogeny shown in Fig. 3.

In the results section, please provide descriptive information regarding the dataset before presenting results from phylogenetic analysis: how many haplotypes, how many segregating or variable sites, how many parsimony informative sites. Also, as I mention before I recommend obtaining a haplotype network.

For phylogenetic analysis and dates of divergence, please explain in the methods why different substitution models were implemented in RAxML and BEAST. Also, in line 226, the authors mention that calibration was constrained to “the time that the oldest known specimen lived (~10.5 kya)”, but it is not clear whether they refer to fossil record or to another type of event, until later were they mention a mummified specimen, please make this clear from the beginning. Also, in the molecular clock analysis, constraining the divergence between the central and norhern clades to 10.5 kya seems inappropiate, as there is no evidence suggesting that the mummified individual used as reference for calibration could belong to either of this lineages and, as fossil record for bats is scarce, divergence between lineages may be older than the specified date. Also, the intervals provided for calibration points should reflect the uncertainty associated to the dating of the fossil record (this would depend on the method use to date the fossil). I recommend, if possible, placing a claibration point at the root of the tree (divergence beteen E. maculatum and outgroup) and let the program estimate the date of divergence between southern, central and northern lineages.

In the discussion section, authors argue that Mexico was the “Pleistocene-era range of E. maculatum”. I suggest also obtaining ENM for LIG period to provide additional evidence supporting this hypothesis. I also suggest adding some points in the discussion regarding the extent of the species in Mexico i.e. does it has a patchy or continuous distribution?

Also, it is not clear from Fig 3. why the southern lineage is defined as paraphyletic as only one sample was included in the analysis and outgroup is not shown.

In the discussion, some extrapolation regarding the possible response of the species to future climate change is made. Nevertheless, this should be taken with caution as no formal niche analyses or future projections were conducted (i.e. extent of the environmental range occupied by the species is not analyzed), and the response of species to climate change will depend not only on dispersal capacity but also in physiological constrains and resource availability (i.e. refuge and food). Even models projected into future climate change have uncertainty associated and results should be interpreted with caution, so I recommend caution in this regard.

Authors mention that all data is available, accordingly they provide accession numbers for genetic data, but there is no information regarding occurrences used to build ENMs and it is not mentioned whether this data was downloaded from GBIF od VertNet or another public repository. If so, please provide de DOI of the dataset.

Line 49. Mention of models for future projections seems a little bit out of context because authors are referring to the usefulness of past projections for the understanding of phylogeography and historical demography. Please clarify.

Line 56. There is a typo in northeastern.

Line 82. When you mention that some there are hotspots where bats were found death or dying. Is there any information regarding the cause of death? Are these bats affected by white nose syndrome?

Lines 124-125. How were rear edge and leading edge defined? Was the centroid for the ENM estimated, as sometimes the niche centroid and geographic centroid may show displacement.

Lines 325-327. Accession numbers should be provided in the methods section.

Lines 339-340. I recommend removing these lines, as evaluating the response of E. maculatum to future climate change is not an objective of this study.

Line 349. Please check wording. Is range extraction correct?

Lines 361-362. High number of substitutions between sample from Queretaro and the central clade may relate to the high geographic distance between sampling sites, as it is not clear if this populations from Mexico are geographically isolated or if species distribution is continues and there is only a lack of information for the Mexican populations.

Lines 364-366. Authors suggest that there is niche conservatism for the last 20,000 years, but it seams that there may be some degree of niche differentiation between the northern clade (inhabiting temperate forests) and the central and southern clades (inhabitin xeric shrublands). Please, revise this sentence. Perhaps an exploratory analysis of environmental data, such as PCA, could shade more light regarding niche conservatism or niche differentiation.

Lines 372-375. Please be careful as not to overinterpret the data, as genetic information from Cuatro Cienegas and Big Bend is not available, the hypothesis of a Pleistocene refugia in this area cannot be tested. In my opinion, authors should suggest obtaining samples from this area, and Mexico in general, is needed to better-understand the evolutionary history of this species. Also, I recommend conducting ENM projections to LIG, as both, ENM and genetic data from Mexico, would provide information on deeper divergence and demographic processes.

Line 481. Please especify which type of preys.

Figure 1. Please depict the site where the mummified specimen was found.

Figure 3. Please improve figure 3, numbers in brackets are unreadable. Was I. phyllotis used as outgroup, as this figure suggest that calibration point was placed in the base of North American lineages, excluding the lineage from Mexico. Please check. Also, the phylogeny shown in top left panel is not clear. Please improve figure.

6. PLOS authors have the option to publish the peer review history of their article (what does this mean?). If published, this will include your full peer review and any attached files.

Reviewer #1: No

Reviewer #2: No

---

## [Author Response · Author response to Decision Letter 0]

10 Mar 2023

Reviewer #1: Review of the article titled “Once upon a time in Mexico: Holocene phylogeography of the spotted bat (Euderma maculatum)” (ID: PONE-D-22-23686). Sanchez et al. present a survey on the biogeography of rare bat species in western North America. The idea of this study is not innovative but it is an interesting and important contribution to the knowledge of past and current distribution and diversity of Nearctic bats, particularly species of xeric environments.

This study is also helpful for proper planning of conservation for the threatened bat species as genetic data show that its protection needs to consider the spatio-genetic structure of populations and niche modeling indicate which areas are crucial for preserving the distribution of this species. However, there are also some problems with this study and some of them are important, therefore my opinion about the publication of this research is ambiguous.

First, as this study is broadly focused on niche modeling and phylogeographic data are additional I would suggest changing the title to “… biogeography … “ or informing that the study is about Holocene phylogeography and past and current niche modeling.

Thank you. We changed the title to more broadly encompass the interdisciplinary methods.

Next, niche modeling is explained (both in methods and results) with many details and I do not see any problems with the use of this method (with the minor question below). I would only ask why the Authors did not include in modeling also other periods – especially prediction of distribution in the future (there are available predictions of climate for at least 2050-2070), what could bring interesting information for the protection of this species.

In l. 149-150 it is informed that modeling was based on climatic data, whereas in l.160 also altitude was added to variables. It is not justified in the paper, however, for me, it is a good decision as this bat species is present mostly in mountainous areas.

Thank you, we appreciate this idea. We added a projection to the last interglacial (~130 kya) to validate that the ancestral range was Mexico. We did not do future projections because the focus of our question is on the past. 

We also clarified that the altitude layer was only used to crop a spatial extent from which to draw background points for training our model. This allowed us to frame model training within the extent of the US Intermountain West while accounting for space that could be occupied, regardless of detection. We wish to clarify that we modeled using climatic data because our question focused on the suitability of climate space and how it changed from the Pleistocene-Holocene transition. We do believe and hope that more sophisticated models can be built that account for terrain in the future, particularly for questions involving a more accurate picture of the contemporary distribution. 

Use of Idionycteris phyllotis is not justified in methods until the end (l. 247) – it should be explained in l. 171 that this species is used as an outgroup. I wonder why this species which belongs to a different genus than Euderma?

Great point. We have added text to explain this. Euderma is a monotypic genus. Idionycteris phyllotis (also a monotypic genus) is the most closely related species to E. maculatum, having diverged ~20 mya and so is the best option for an outgroup.

I wonder why the Authors used only a single genetic marker for phylogeography? CO1 is widely used as a barcode and marker in phylogeography, however, its utility is limited because of the mode of its inheritance. Moreover, single locus analyses could not solve the whole history and contemporary diversity of the species. It is not justified in the text why not to use other markers like some introns, microsatellites or genomic data. Especially SNPs based on NGS are now standard in any phylogenetic or phylogeographic studies. In my opinion, the use of only mtDNA is outdated and should no be longer accepted in research, maybe just in some preliminary works.

We used two mtDNA markers (1 locus): D-loop and cyt B. We did not use COI. As a single locus, mtDNA provides a partial assessment of genetic diversity; we agree that nuclear introns and certainly genomic SNP data would add a more complete picture. Our lab is currently using nuDNA to genotype E. maculatum but this still requires a year or two of work. We believe our ecological niche model (ENM) findings are important to share now and mtDNA helps provide context for the patterns we observed in the ENM. This structure can provide an argument for a priori groupings for population genetic analysis (e.g., microsatellites, SNPs) and identifies haplotypes, which can be useful in building SNP panels to avoid ascertainment bias. While it is true that mtDNA only resolves a single-locus gene tree, its mode of inheritance also makes it very sensitive to signals of genetic drift, which makes it well suited for identifying colonization history throughout the last 10,000 years.

Next, for the only Mexican site available only a single individual, moreover old museum sample. I wonder if the Authors managed to sequence this sample without problems and if the reads were of high quality to be sure that all polymorphisms are real, especially since this was only one sample so there was no reference. I am pretty sure that these individuals belong to different lineages but it is not appropriate to draw conclusions from a single sample.

Thanks, this is a great point. Yes, these sequences were of good quality and the sample has been sequenced multiple times. We were able to add the second Querétaro specimen to our genetic dataset, which gave the same haplotype as the other. We also added a specimen from Chihuahua, Texas, and California but were only able to sequence using D-loop in the time we had. 

Regarding analyses. This is mostly phylogeographic study, whereas methods in use are more phylogenetic (tree reconstruction, dating). I wonder why the Authors did not use e.g. networks, some statistics describing current diversity (AMOVA, IBD etc.) or gene flow (simply Fst or some coalescent methods) or demography (Mismatch Distribution or skyline plots). Moreover, as 2 (or 3) distinct units were identified it would be interesting to test if they should be considered as distinct taxa, I mean especially Mexican bat (but again – it is only a single sample). 

Great idea. We added a haplotype network, sequence diversity summaries, extended Bayesian Skyline analysis, and neutrality tests. We don’t have appropriate sample size, marker selection and geographic coverage to study gene flow, nor justifiable groupings for contemporary structure (AMOVA, Fst, IBD). Use of mtDNA is not an appropriate data type for isolation by distance and would be better addressed with nuDNA. However, we are currently putting together a nuDNA study to address questions related to these analyses. We do not intend for this to be a systematics study but we do hope that our current findings can aid future studies in determining if the Mexico lineage we identified could be a distinct species (if it still occurs in the area).

About dating, I am not sure what was the reason to assume that the mummy represents the moment of split between central and northern lineages. Why its age could not be associated with earlier periods (before this split) or later periods (being members of either the central or northern clade)? Are there any morphological features of this mummy which suggest the assignment or it was based on some other data (paleontological?)?

The placement of the fossil is an important question that required a lot of discussion in the planning phase. However, based on running the phylogenetic analysis with multiple methods and getting incongruent topologies, we have omitted divergence dating. We replaced this analysis with a Bayesian skyline analysis with a clock rate from a near neighbor to evaluate timing of demographic pattern.

Moreover, it is possible that in Mexico exist other lineages of this bat and more samples from various localities are needed to fully understand its phylogenetic diversity. Some of the conclusions of this study are uncertain due to that tiny sampling e.g. l. 360 about paraphyletic of S lineage – it could not be confirmed without more samples as other samples from northern Mexico could be intermediate. Generally, I am not sure if paraphyletic is an appropriate term here as all 3 lineages form a monophyletic group and members of the central and northern lineages belong to single clades so are also monophyletic. Southern lineage is just one sequence so its phylogenetic relationship to other lineages could change with more data. I am not sure also in Mexico could be assigned as an ancestral range as members of other lineages are absent in Mexico (at least based on available data), so their ancestral areas could be rather in SW USA.

Good point. We agree, and have dialed back on phylogenetic interpretations, and focused more on haplotypic interpretations. We acknowledge in text that the species is data deficient because samples are difficult to come by. Based on the haplotype network we were able to include in our revisions, northern Mexico is marginally intermediate and more related to the central lineage.

To the reviewer’s point, a lack of samples in the Mexican range is likely a sampling bias. Most of the heavy work for this species was done between the 1970s – 1990s in the US and Canada. Very little work has been done in Mexico and we think our findings will galvanize future efforts in that portion of the range. We recently became aware that contemporary trapping efforts in the area have come up unsuccessful. We are happy to verify this information with a personal correspondence if the editor finds this information valuable to our study.

Reviewer #2: Sánchez et al. conducted ENM and phylogenetic analyses to assess how post-glacial climate changes influenced the phylogeographic structure of Euderma maculatum. The manuscript is well written and interesting. There are some points that the authors should address and clarify.

In introduction, please clarify in “we asked whether maternal genetic data could better resolve phylogeographic structure”. This sentence suggests that there were previous unsuccessful attempts to resolve the phylogeographic structure of this species. Please clarify, and if there was a previous study provide background information about it.

We have removed the word “better”. This was mentioned in text as a study (Best 1988, citation 37) that used morphological characteristics for hierarchical clustering. Because this study used a priori, population-based groupings, this source did not provide a valid phylogenetic pattern.

In methods section please mention the extension to which the bioclimatic layers were cropped for ENM analyses. Also, mention the AOGCM used for past projections, and justify why not evaluating range dynamics to the LIG. In addition, it is not clear which database was searched to obtain occurrence points.

We have visualized the cropped layer used for training and made available in Supplementary Data. We also included AOGCMs. We appreciated your idea to project to LIG and did so.

We clarified that we used Vertnet, Arctos, and field captures to locate specimens and occurrence points and have clarified in text. We have provided these data in supplementary files.

In the introduction the authors mention that E. maculatum originated in Mexico; nevertheless, only one sample from Mexico was included in the study. In a quick search in GBIF I notice that there are additional occurrence points from Mexico, was this database searched or were these points excluded for some reason? Please, in methods section, justify or explain why only one point from Mexico was used to obtain ENM. Moreover, in the discussion section, mention the limitations of lacking samples from the center of origin (where ancestral haplotypes are expected) in the ENM and phylogeographic analyses. Particularly as a lack of information is not the same as a true absence, and this cause bias in ENM models and past projections.

Thank you. As mentioned above, we used Vertnet, Arctos, and our team’s own records of field capture to acquire geographic occurrences and specimens. We have included these occurrence data as well as the dereplicated points in the supplementary material. For genetic analysis, we added another Querétaro sample as well as two specimens from northern Mexico. However, we were only able to sequence it at D-loop in the time we had. Because Vertnet did not have information for the Durango localities, we overlooked it when collecting occurrence data for our ENM. However, we do not believe that excluding this locality substantially affected the model. In support of this, we recently became aware of an emerging ENM project from another group who used occurrence data from GBIF (presumably includes the Durango occurrences) and found identical results regarding hindcasting. This study was recently presented at NASBR/IBRC Joint Symposium 2022 entitled “Niche Tracking of Dry Conditions in the Spotted Bat (Euderma maculatum)” by Camilo Calderón Acevedo.

We did not use the Querétaro occurrences in the model because the coordinates were approximated in their source publications and were therefore unreliable for modeling. We did use two Chihuahua, MEX occurrences, but because they were from the same location they were consequently dereplicated to a single occurrence point. It’s important to note that GBIF has many duplicated occurrences of some of the same specimens have different coordinates. Additionally, many of these occurrences in GBIF lacked sources for context.

In text, we mentioned limitations of lacking samples, particularly from Durango. We have specifically noted that the context of our model was more of the central range. However, we believe the model provides a sufficient picture of where we know the species to occur. Based on how the model was trained, it impressively predicted Mexican range but only when hindcasted.

Apparently, there is some correspondence between the northern and central lineages and habitat type (temperate forest and xeric shrubland; but not sure as habitat type is not depicted in Fig. 3); If so, it would be interesting to test for niche differentiation between lineages.

Thank you for this suggestion. In person, the habitat types look almost the same in the north as they do in the southwest (but of course this will vary based on scale). A literature search suggests that there are papers with guidance for niche differentiation but it is clear that this is not a minor endeavor. We respectfully think that this would be more appropriate for another study that focuses primarily on the question of niche differentiation.

Also, please provide a haplotype network, as this type of analysis depicts genealogical relationships among haplotypes and also could provide some clues regarding the historical demography of this lineages, as we could expect to find signals of population expansion that are consistent with the range expansion mentioned in the discussion.

Thank you. We included a haplotype network and analyses of historical demography.

For the organization of the manuscript, I suggest moving the corroboration of the calibration assumptions before presenting the final phylogeny as this analysis provide support to the phylogeny shown in Fig. 3.

After re-evaluating the topologies and finding rooting incongruencies with additional methods, we omitted phylogenetic divergence dating and instead focused on demographic histories, calibrated using published clock rates from near neighbors.

In the results section, please provide descriptive information regarding the dataset before presenting results from phylogenetic analysis: how many haplotypes, how many segregating or variable sites, how many parsimony informative sites. Also, as I mention before I recommend obtaining a haplotype network.

We now provide sequence/haplotype diversity estimates with those metrics and a haplotype network.

For phylogenetic analysis and dates of divergence, please explain in the methods why different substitution models were implemented in RAxML and BEAST. Also, in line 226, the authors mention that calibration was constrained to “the time that the oldest known specimen lived (~10.5 kya)”, but it is not clear whether they refer to fossil record or to another type of event, until later were they mention a mummified specimen, please make this clear from the beginning. Also, in the molecular clock analysis, constraining the divergence between the central and norhern clades to 10.5 kya seems inappropiate, as there is no evidence suggesting that the mummified individual used as reference for calibration could belong to either of this lineages and, as fossil record for bats is scarce, divergence between lineages may be older than the specified date. Also, the intervals provided for calibration points should reflect the uncertainty associated to the dating of the fossil record (this would depend on the method use to date the fossil). I recommend, if possible, placing a claibration point at the root of the tree (divergence beteen E. maculatum and outgroup) and let the program estimate the date of divergence between southern, central and northern lineages.

We decided to re-evaluate the divergence dating design and agree that our fossil placement was not justified. We decided to look at other tree methods and found inconsistencies in rooting topologies. Therefore, we found phylogenetic divergence dating unreliable for this marker and sample set. We instead used a Bayesian Skyline analysis to infer historic patterns of demography.

In the discussion section, authors argue that Mexico was the “Pleistocene-era range of E. maculatum”. I suggest also obtaining ENM for LIG period to provide additional evidence supporting this hypothesis. I also suggest adding some points in the discussion regarding the extent of the species in Mexico i.e. does it has a patchy or continuous distribution?

Great idea. We projected to the LIG and found that most of the distribution was still largely in Mexico. The overall range is known to be patchy. The extent of the Mexican distribution is unknown and given the lack of observations after the 1980s and recent correspondence with a leading bat researcher in the area, may no longer exist due to a lack of detections. We hope our work will inspire future efforts to better survey the region.

Also, it is not clear from Fig 3. why the southern lineage is defined as paraphyletic as only one sample was included in the analysis and outgroup is not shown.

We thank you for this observation and we have recognized our mistake. We no longer anchor our interpretations on phylogenetic pattern. Instead, we now rely on the ENM and haplotype networks to support this claim.

In the discussion, some extrapolation regarding the possible response of the species to future climate change is made. Nevertheless, this should be taken with caution as no formal niche analyses or future projections were conducted (i.e. extent of the environmental range occupied by the species is not analyzed), and the response of species to climate change will depend not only on dispersal capacity but also in physiological constrains and resource availability (i.e. refuge and food). Even models projected into future climate change have uncertainty associated and results should be interpreted with caution, so I recommend caution in this regard.

Thank you, we devoted an entire paragraph to this.

Authors mention that all data is available, accordingly they provide accession numbers for genetic data, but there is no information regarding occurrences used to build ENMs and it is not mentioned whether this data was downloaded from GBIF od VertNet or another public repository. If so, please provide de DOI of the dataset.

We now have included this information in supplementary files.

Line 49. Mention of models for future projections seems a little bit out of context because authors are referring to the usefulness of past projections for the understanding of phylogeography and historical demography. Please clarify.

Thank you, we have provided more context. We go on in the next sentence to clarify that the past is important for understanding the future.

Line 56. There is a typo in northeastern.

Fixed

Line 82. When you mention that some there are hotspots where bats were found death or dying. Is there any information regarding the cause of death? Are these bats affected by white nose syndrome?

Most individuals are found and studied in hotspots because these sites are more likely to allow for captures. Even in hotspots, they are still quite difficult to capture, representing 1% of captures. Generally, from reports of dead or dying individuals, exhaustion is the cause of death (long flights). We have added information to the methods where we discuss occurrence criteria. To our knowledge, WNS has not been observed for this species. This bat is not suspected to be affected by WNS because they primarily roost solitarily or in low densities in sheer cliff walls. The greatest conservation concern for the species is data deficiency.

Lines 124-125. How were rear edge and leading edge defined? Was the centroid for the ENM estimated, as sometimes the niche centroid and geographic centroid may show displacement.

We did not estimate niche or geographic centroid. We simply defined based on northern-most occurrences, supported by a distinct haplogroup and southern-most population and its unique haplotype.

Lines 325-327. Accession numbers should be provided in the methods section.

We have moved the mention of accession numbers to Methods section, following sequencing methods.

Lines 339-340. I recommend removing these lines, as evaluating the response of E. maculatum to future climate change is not an objective of this study.

We have deleted these lines.

Line 349. Please check wording. Is range extraction correct?

Typo. We changed to expansion.

Lines 361-362. High number of substitutions between sample from Queretaro and the central clade may relate to the high geographic distance between sampling sites, as it is not clear if this populations from Mexico are geographically isolated or if species distribution is continues and there is only a lack of information for the Mexican populations.

Valid point. We added a second Querétaro specimen as well as northern Mexico and Texas specimens into a haplotype network. The species’ distribution is known to be patchy across its range. We have acknowledged lack of available specimens throughout Mexico, and that specimens from Durango would provide more insight. The leading bat biologist for Mexico, Dr. Rodrigo Medellin, has failed to capture them and speculates that they might only remain in Copper Canyon in Mexico. We hope that our work will motivate further sampling in the region given the genetic structure and ENM patterns. The niche model, in particular, supports an idea that they are separated because it not only predicted central Mexican range in the Pleistocene but shows that Mexican climate space has been contracting for the last 130 thousand years. As part of our revision, we added the additional specimens from Texas and Mexico, demonstrating that genetic contiguity (if any) appears to end at Texas and Northern Mexico.

Lines 364-366. Authors suggest that there is niche conservatism for the last 20,000 years, but it seams that there may be some degree of niche differentiation between the northern clade (inhabiting temperate forests) and the central and southern clades (inhabitin xeric shrublands). Please, revise this sentence. Perhaps an exploratory analysis of environmental data, such as PCA, could shade more light regarding niche conservatism or niche differentiation.

We have omitted mention of niche conservatism. This would be an interesting analysis to do in the future.

Lines 372-375. Please be careful as not to overinterpret the data, as genetic information from Cuatro Cienegas and Big Bend is not available, the hypothesis of a Pleistocene refugia in this area cannot be tested. In my opinion, authors should suggest obtaining samples from this area, and Mexico in general, is needed to better-understand the evolutionary history of this species. Also, I recommend conducting ENM projections to LIG, as both, ENM and genetic data from Mexico, would provide information on deeper divergence and demographic processes.

We have removed this interpretation.

Line 481. Please especify which type of preys.

We specified the prey type and included sources. We added more detail in earlier sentences.

Figure 1. Please depict the site where the mummified specimen was found.

We depicted the general location of the fossil.

Figure 3. Please improve figure 3, numbers in brackets are unreadable. Was I. phyllotis used as outgroup, as this figure suggest that calibration point was placed in the base of North American lineages, excluding the lineage from Mexico. Please check. Also, the phylogeny shown in top left panel is not clear. Please improve figure.

We changed this figure based on re-analysis and no longer require numeric annotations.

---

## [Decision Letter · Decision Letter 1]

27 Apr 2023

Once upon a time in Mexico: Holocene biogeography of the spotted bat (*Euderma maculatum*)

PONE-D-22-23686R1

Dear Dr. Sanchez,

We’re pleased to inform you that your manuscript has been judged scientifically suitable for publication and will be formally accepted for publication once it meets all outstanding technical requirements.

Kind regards,

Daniel de Paiva Silva, Ph.D.

Academic Editor

PLOS ONE

Reviewers' comments:

Reviewer's Responses to Questions

**Comments to the Author**

1. If the authors have adequately addressed your comments raised in a previous round of review and you feel that this manuscript is now acceptable for publication, you may indicate that here to bypass the “Comments to the Author” section, enter your conflict of interest statement in the “Confidential to Editor” section, and submit your "Accept" recommendation.

Reviewer #2: All comments have been addressed

Reviewer #3: All comments have been addressed

2. Is the manuscript technically sound, and do the data support the conclusions?

Reviewer #2: Yes

Reviewer #3: Yes

3. Has the statistical analysis been performed appropriately and rigorously? 

Reviewer #2: Yes

Reviewer #3: Yes

4. Have the authors made all data underlying the findings in their manuscript fully available?

Reviewer #2: Yes

Reviewer #3: Yes

5. Is the manuscript presented in an intelligible fashion and written in standard English?

Reviewer #2: Yes

Reviewer #3: Yes

6. Review Comments to the Author

Reviewer #2: (No Response)

Reviewer #3: I have read the revised version of the article titled "Once Upon a Time in Mexico: Holocene Phylogeography of the Spotted Bat (Euderma maculatum)" (ID: PONE-D-22-23686). In general, Sanchez et al. present a comprehensive survey on the biogeography of a rare bat species in western North America, including acoustic surveys, which is great! All the points raised by reviewers 1 and 2 have been completely clarified, including new analyses and data. The manuscript is well-written and interesting. Therefore, in my opinion, the publication is suitable for publication. An additional idea is that the author could use the procedures pointed out by Pimenta et al. (2022) to provide the niche conservatism for the species."

Pimenta et al. (2022) - https://linkinghub.elsevier.com/retrieve/pii/S0304380022001247

7. PLOS authors have the option to publish the peer review history of their article (what does this mean?). If published, this will include your full peer review and any attached files.

Reviewer #2: **Yes: **Gabriela Castellanos-Morales

Reviewer #3: **Yes: **Thiago Bernardi Vieira

---

## [Editor Report · Acceptance letter]

2 May 2023

PONE-D-22-23686R1 

Once upon a time in Mexico: Holocene biogeography of the spotted bat (*Euderma maculatum*) 

Dear Dr. Sanchez:

I'm pleased to inform you that your manuscript has been deemed suitable for publication in PLOS ONE. Congratulations! Your manuscript is now with our production department. 

Kind regards, 

on behalf of

Dr. Daniel de Paiva Silva 

Academic Editor

PLOS ONE